



# 1 Reconstructed glacier area and volume changes in the European Alps
# 2 since the Little Ice Age

Johannes Reinthaler[1], Frank Paul[1]
[1]Department of Geography, University of Zurich, Zurich, Switzerland
*Correspondence to*: Johannes Reinthaler (johannes.reinthaler@geo.uzh.ch)
**Abstract.** Glaciers in the European Alps have experienced drastic area and volume loss since the end of the Little Ice Age
(LIA) around the year 1850. How large these losses were is only poorly known as published estimates of area loss are mostly
based on simple up-scaling and alpine-wide reconstructions of LIA glacier surfaces are lacking. For this study, we compiled
all digitally available LIA glacier extents for the Alps and added missing outlines for glaciers >0.1 km$^2$ by manual digitising.
This was based on geomorphologic interpretation of moraines and trimlines on very high-resolution images in combination
with historic topographic maps and modern glacier outlines. Glacier area changes are determined for all glaciers with LIA
extents at a regional scale. Glacier surface reconstruction with a Geographic Information System (GIS) was applied to calculate
(a) glacier volume changes for the entire region from the LIA until today and (b) total LIA glacier volume in combination with
a reconstructed glacier bed. The glacier area shrunk by 2405 km$^2$ (-57%) from 4211 km$^2$ at the LIA maximum to 1806 km$^2$ in
2015 and volume was reduced from about 281 km$^3$ around 1850 to 100 km$^3$ (-65%) in 2015, roughly in line with previous
estimates. In the mean, glacier surfaces lowered by -43.3 m until 2015 (-0.25 m a$^{-1}$), which is three-times less than observed
over the 2000 to 2015 period (-0.82 m a$^{-1}$). The strongest volume losses occurred at around 1600 m and at least 1832 glaciers
melted away completely. Many glaciers have now only remnants of their former coverage left, which led to deglaciation of
entire catchments. The new datasets should support a wide range of studies related to the effects of climate change in the Alps.

## 21 1 Introduction

Glaciers in the European Alps are among the most intensely studied worldwide. During recent decades, increasing temperatures
caused accelerated glacier retreat and downwasting, impacting water supplies during dry periods, glacier forefield ecosystems,
slope stability and tourism (Brunner et al., 2019; Cannone et al., 2008; Haeberli et al., 2007; Oppikofer et al., 2008). While it
is crucial to determine the future evolution of glaciers and its consequences, reconstructing past glacier extents and changes
allows us to put possible future developments into perspective. Direct observation of glacier extents (including pictorial
evidence) and first measurements of front variations in the Alps date back to pre-industrial times (e.g. Zumbühl and
Nussbaumer (2018), whereas first topographic maps with glacier extents were published in the 19[th] century for different Alpine
regions (Table S1). The large body of literature presenting outlines from historic glacier extents in the Alps and elsewhere (e.g.
Grove, 2001) in printed (analogue) form is hard to use in today's digital world and need to be digitized and geocoded first.



As an alternative, very high-resolution satellite or aerial images allow us to identify and delineate terminal and lateral moraines
or trim lines to reconstruct Little Ice Age (LIA) glacier extents (e.g. Reinthaler and Paul, 2023; Lee et al., 2021). For the Alps,
numerous studies have already created LIA inventories for specific regions or countries (e.g. Colucci and Žebre, 2016; Fischer
et al., 2015; Garent and Deline, 2011; Knoll et al., 2009; Lucchesi et al., 2014; Maisch et al., 2000; Nigrelli et al., 2015;
Zanoner et al., 2017) and the related outlines are freely available from open repositories or the Global Land Ice Measurements
from Space (GLIMS) glacier database (Table S2). However, for some regions in the Alps (19% of all glaciers according to the
Randolph Glacier Inventory, RGI v7.0) digitized LIA glacier extents were not available and have been newly digitised in this
study (Figure 1, Table S3).
In the Alps, several glacier maximum extents occurred between 1350 and 1850/60, the exact timing depending on the glacier
(e.g. Zumbühl and Holzhauser, 1988; Nussbaumer et al., 2011). From these, only the moraines and trimlines from the last
maximum (around 1850) are sufficiently complete for digitizing. However, in contrast to other regions in the world, extent
differences (e.g. between 1850, 1820 or 1600) are small in the Alps. For example, most glaciers in the Italian and western Alps
reached their last maximum extent around 1820, but re-advanced to almost the same position around 1850 (Solomina et al.,
2015). In contrast, Austrian glaciers reached their last maximum in the 1850s to 1860s (Ivy-Ochs et al., 2009). Later re-
advances took place in the 1890s, 1920s and 1970s to 80s. Terminal and partly also lateral moraines from these re-advances
can still be seen in several glacier forcefields.
The study by Zemp et al. (2008) suggests a glacier area reduction of almost 50% between 1850 (4474 km$^2$) and 2000
(2271.6 km$^2$) using a size-dependent extrapolation scheme to obtain alpine-wide extents for 1850. The study by Hoelzle et al.
(2003) used parameterisation schemes (e.g. to derive mass balance from length changes) whereas Colucci and Žebre (2016)
used volume-area scaling to derive former glacier volume for the Julian Alps. By reconstructing the former glacier surface
directly, distributed glacier thicknesses and elevation changes can be derived. Whereas reconstructions of glacier extent and
surfaces for the LIA maximum have been compiled and published for many regions around the world, for example Patagonia
by Glasser et al. (2011), Greenlands peripheral glaciers by Carrivick et al. (2023), Himalaya by Lee et al. (2021) and New
Zealand by Carrivick et al. (2020), this information was so far not available for the entire European Alps.
This study presents a first compilation of LIA maximum glacier extents along with a reconstruction of their surfaces and
calculation of their volumes for all glaciers in the Alps larger than 0.1 km$^2$. Furthermore, we quantify changes in glacier area,
volume and elevation between the LIA and around the year 2000 and analyse related spatial variations at the regional scale.





## 2    Datasets and Methods

### 2.1    Regional subdivision

For the regional-scale calculations, we have adopted the International Standardized Mountain Subdivision of the Alps (Marazzi, 2004), which was previously also used by Sommer et al. (2020) to aggregate recent glacier mass changes. The dataset consists of two main subdivisions into the Eastern and Western Alps and 14 subdivisions into smaller regions (see Figure 1). Regions with a very small glacier coverage (<5 km$^2$) were merged with neighbouring regions (Maritime with Cottian Alps; German Prealps with Austrian Prealps). Glacier area and volume changes were also calculated per country and for five major river basins (Rhine, Rhone, Danube, Po, SE Alps).

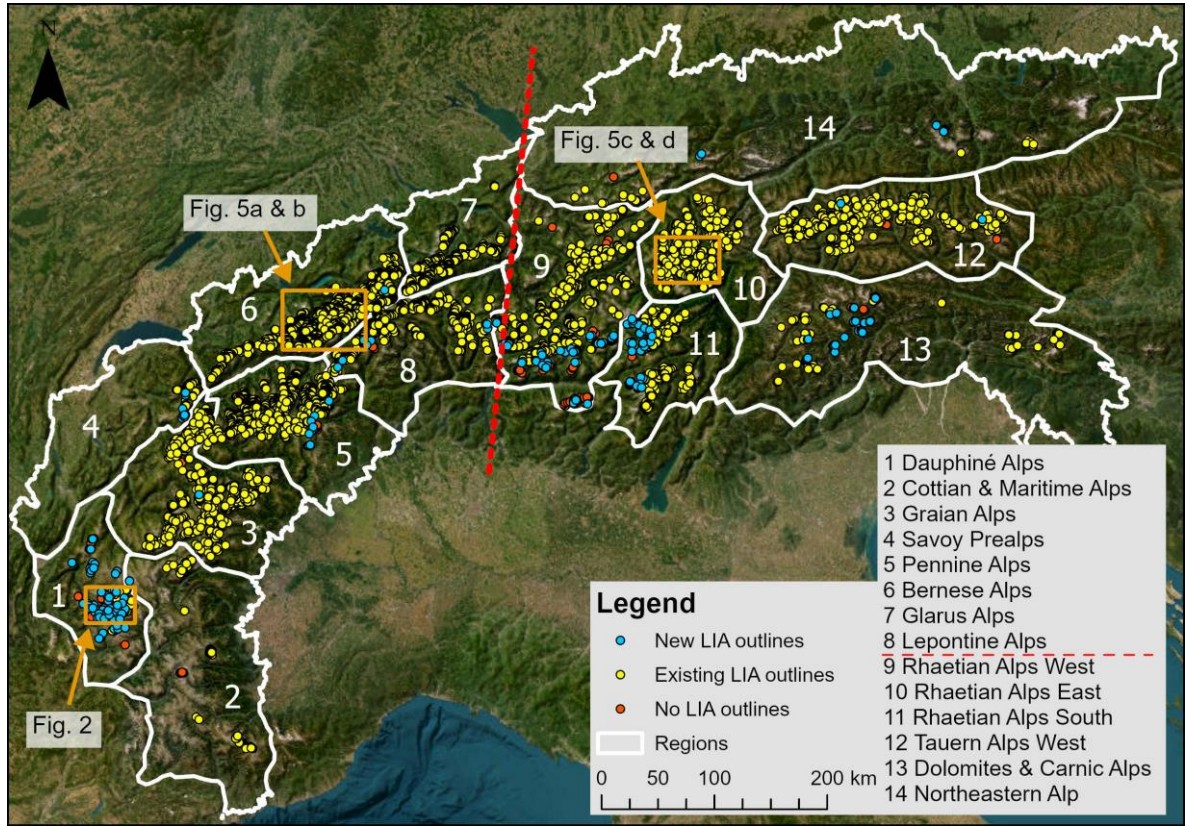

**Figure 1: Study region of the European Alps. In white are the 14 sub-regions, in yellow the existing LIA glacier outlines (3891 km$^2$) from various sources (see Table S2), in blue the new LIA glacier outlines (329 km$^2$) and in red the glaciers of the RGI v7.0 (<0.1 km$^2$) without a LIA equivalent (6.8 km$^2$). The orange squares denote the location of sub-regions shown in Figs. 2 and 5. The red dashed line marks the division between Eastern and Western Alps. Background image: ESRI, (2023b).**

### 2.2    Glacier outlines

We have used glacier outlines representing maximum LIA extents from various sources (see Table S2), a second dataset from the year 2003 compiled by Paul et al. (2011) and available from RGI v7.0 and a third dataset from around 2015/16 described by Paul et al. (2020) and available from GLIMS. The 2003 inventory has been derived from Landsat 5 images and the 2015/16



inventory from Sentinel-2; both datasets were taken as they are and not modified. Due to differences in interpretation of glacier
extents by different analysts for the two datasets, we will only present glacier changes at a regional scale rather than per glacier.
Missing LIA extents were digitized for important individual glaciers and glaciers larger than 0.1 km$^2$ in RGI v7.0 based on the
geomorphological interpretation of trimlines and frontal as well as lateral moraines as visible on very high-resolution images
(Figure 2). These images were provided by web map services from ESRI (world imagery, standard and clarity (ESRI, 2023b),
Google (https://earth.google.com/web/) and Bing (www.bing.com/maps) and used in combination with roughly geocoded
historical maps (see Table S1 for details) to aid in the interpretation. As a starting point for the LIA outline digitizing, we used
outlines from 1967-1971 (for France) and the RGI v7.0 from 2003 for the other regions.

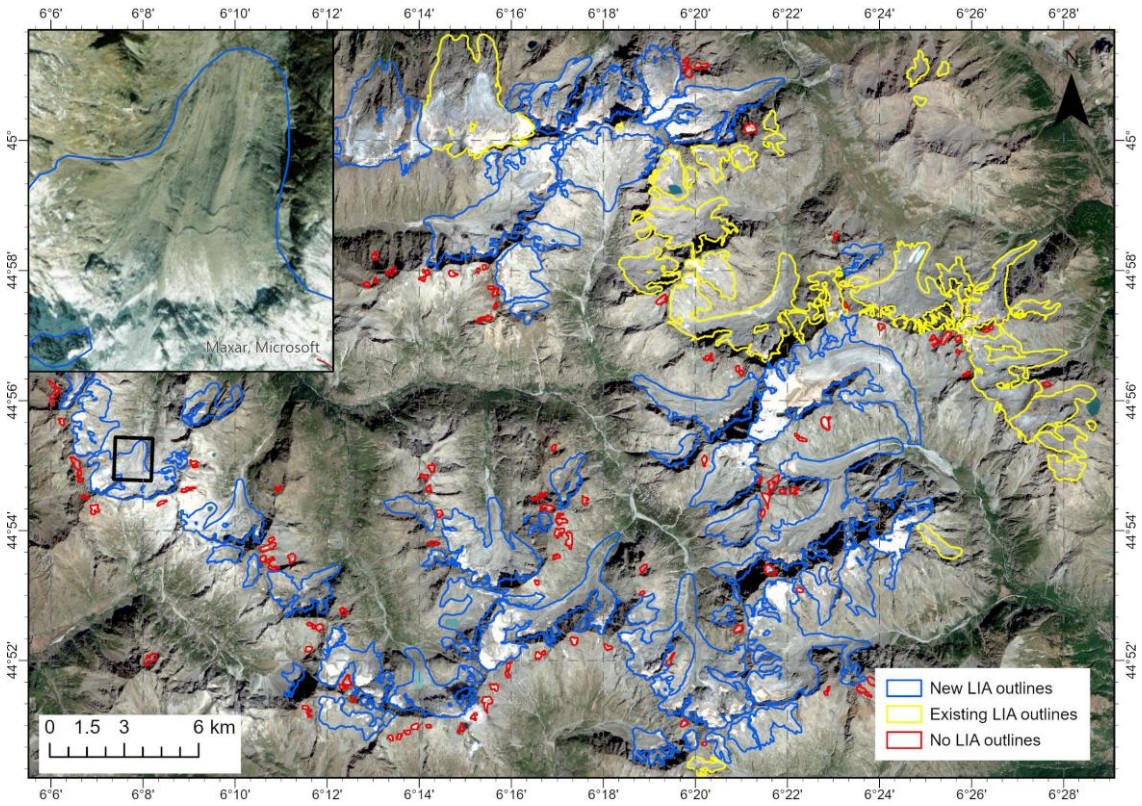


**Figure 2: For the example region of the Dauphiné Alps we show the new (blue) and existing (yellow) LIA outlines as well as glaciers**
**smaller than 0.1 km$^2$ without LIA outlines (red). The black box shows the location of the inset above it, illustrating multiple recession**
**moraines and the resulting LIA outline. Background image: Sentinel-2, acquired on 12.08.2022, source: Copernicus Sentinel data**
**2022.**
The largest regions without published LIA outlines were the Italian region of Lombardy and parts of the Dauphiné Alps (see
Table S3 for a list of regions with previously missing LIA glacier extents). For the glaciers in Germany, published maps
(Hirtlreiter, 1992) were combined with late 19$^{th}$-century outlines (available from www.bayerische-gletscher.de) and extended
to visible moraines. In total, around 767 glaciers (in RGI v7.0) did not have a LIA equivalent of which 389 now have one (216
glaciers at LIA). The remaining unconsidered glaciers are small (<0.1 km$^2$) and are not expected to change the area and volume





change calculation on a regional scale substantially (they have a total area of 12.1 km$^2$) when neglecting them. The existing
and new LIA datasets combined cover 99.4% of the 2003 glacier area in RGI v7.0 ( Figure 1). A few glaciers that melted away
before 2003 would lower this number by a few decimals.

## 2.3    GIS-based surface reconstruction

The reconstruction of glacier surfaces is based on elevation information along the LIA outlines and interpolation of the area in
between. The elevation was extracted from the 10 m resolution Copernicus DEM acquired by TanDEM-X between 2011 and
2015 (ESA, 2019) along points on the outlines with 100 m equidistance. The interpolation is based on the up-scaling approach
presented by Reinthaler and Paul (in review.) that uses the pattern of recent elevation change rates for each glacier (Hugonnet
et al., 2021) to calculate glacier-specific elevation change gradients from bilinear interpolation through all points. The method
calculates a scaling factor by dividing the gradient by the LIA elevation change (from interpolating outline points only). The
resulting scaling factor (median per region; Figure S1) is then applied to the gradient to shift the modern DEM to the LIA
elevation of the glacier. The surface for the area between the modern and the LIA outline was interpolated using Topo to Raster
(optimised for point input data without enforcing drainage). For glaciers where no relationship between elevation change and
elevation was found i.e. no elevation change gradient, only the outline points were interpolated. The result is a 30 m resolution
DEM of LIA glacier surfaces for nearly all glaciers in the Alps. From this DEM, topographic properties (e.g. median, minimum
elevation, slope) were extracted for each glacier.

## 2.4    Volume reconstruction and change assessment

In Table 1, the raster datasets used for the volume reconstruction and change assessment are listed. Calibrated glacier bed
datasets were used to calculate the total glacier volume for Switzerland (Grab et al., 2021) and Austria (Helfricht et al., 2019).
For the remaining regions, the glacier bed that was inverted from glacier thickness data by Millan et al. (2022) combined with
the Copernicus DEM was used to determine glacier volume. All glacier bed datasets were clipped to the 2015 glacier extent.
Area, elevation and volume changes were calculated since the LIA until around the year 2000 (DEM from 2000, outlines from
2003) and around 2015 (change rates from Hugonnet et al. (2021) between 2000 and 2014, DEM and outlines from 2015/16).
To simplify the presentation of changes, we refer to the time periods P1 (LIA-2000), P2 (2000-2015) and P3 (LIA-2015) even
though outlines, DEMs and change rates refer to slightly different years. Similarly, we have used the year 1850 as the date of
maximum LIA extent, even though individual glaciers may have reached their maximum extent at different times. Glacier
changes for time periods between the LIA and 2000 were quantified by Helfricht et al. (2019) for Austria and Mannerfelt et
al. (2022) for Switzerland but so far not for the entire European Alps. As these studies used different input datasets (outlines,
DEMs) as a base and refer to different time periods, we do not compare our results with results from these studies.

The void-filled SRTM DEM (3-arc) and the Copernicus DEM were used as the year 2000 and 2015 glacier surfaces to calculate
volume and elevation changes for P1 and P3, respectively (ESA, 2019; NASA Shuttle Radar Topography Mission (SRTM),





2013). Elevation change rates for P2 were taken from Hugonnet et al. (2021) as it is widely used and because issues with radar
penetration of the SRTM and Copernicus DEMs are much more prominent over the much shorter time period (e.g. Dehecq et
al. 2016). This probably resulted in positive elevation changes in several accumulation areas (Figure S6).

**Table 1: Raster datasets used for the glacier change assessment.**

| Dataset | Reference | Region | Used for | Date | Time period |
|---|---|---|---|---|---|
| LIA surface DEM | This study | Alps | Volume/elevation change (rate) | LIA (1850) | P1 & P3 |
| Copernicus DEM | ESA, 2019 | Alps | Volume/elevation change (rate) | 2011-2015 | P3 |
| SRTM DEM | NASA Shuttle Radar Topography Mission (SRTM), 2013 | Alps | Volume/elevation change (rate) | 2000 | P1 |
| Elevation change rate | Hugonnet et al., 2021 | Alps | Volume/elevation change (rate) | 2000-2014 | P2 |
| Glacier bed | Grab et al., 2021 | Switzerland | Total glacier volume | N/A | |
| Glacier bed | Helfricht et al., 2019 | Austria | Total glacier volume | N/A | |
| Glacier thickness | Millan et al. 2022 | Alps except Austria and Switzerland | Total glacier volume | 2017-2018 | |


**2.5    Uncertainty assessment**
We applied a simplified approach to quantify all relevant sources of uncertainty on the total LIA volume and volume changes
rather than a glacier or cell-specific uncertainty assessment as used by Martín-Español et al. (2016). The main reasons are our
highly variable input datasets and methods as well as the focus on regional rather than glacier-specific changes.
The overall area uncertainty of the digitized LIA glacier outlines is about ±5% (Reinthaler and Paul, 2023), but the relative
area uncertainty is lower for larger and higher for smaller glaciers (Paul et al., 2013). Due to a lack of reference data, an
uncertainty assessment of the reconstructed LIA surfaces is difficult. However, for a case study in the Bernese Alps, the mean
difference to a dataset derived by Paul (2010) from digitised historic contour lines with 100 m equidistance could be obtained
and was quantified to 4.6 m (Reinthaler and Paul, in review.), which gives an uncertainty of the total volume of 6.9%.
Uncertainties of the bed topography impact on the ice volume are in the range of 4.1 to 5% for the calibrated (Grab et al., 2021;
Helfricht et al., 2019) and up to 30% for the un-calibrated datasets (Millan et al., 2022). Considering the proportions of the
three datasets, the overall uncertainty regarding the bed topography was quantified to 12.7% (details in supplement).
Combining the uncertainty relating to glacier outlines, surface reconstruction and bed topography, the total random error of
the glacier volume is calculated as $\varepsilon = \sqrt{(5.05^2 + 6.9^2 + 12.7^2)}$ or 15.3%.
Excluding the glaciers without a reconstructed extent and the missing glaciers leads to a systematic underestimation of the
volume and volume change calculations, i.e. this introduces a bias. For the already existing LIA outline datasets, almost all
LIA glacier extents were digitised in the related studies (independent of their size), including those that have since melted
away. For the glaciers >0.1 km$^2$ in RGI v7.0 that do not have LIA extents (total area of 12.1 km$^2$), we have extrapolated their





LIA area from the mean relative change of the size class smaller than 1 km$^2$ to 38.37 km$^2$ with an estimated total volume of
0.26 km$^3$ when using the parameterisation scheme by Haeberli and Hoelzle (1995) and a constant mean ice thickness. For
already disappeared glaciers that were not mapped, the quantification of their area and volume is more challenging. According
to Parkes and Marzeion (2018), disappeared glaciers globally accounted for 4.4 mm (lower bound) of SLR compared to
89.1 mm for all glaciers in RGI v5.0 (4.9%). Using the lower bound, since many glaciers disappeared were mapped in the
Alps, this would lead to a total underestimation of the volume of around 14.1 km$^3$ (5.0%). For the volumes of 2003 and 2015,
only the uncertainty regarding the bed topography is considered, but DEM accuracy and glacier mapping uncertainties add to
the overall uncertainty.

## 3 Results

### 3.1 Glacier area changes

The total LIA glacier area of the Alps was reconstructed to 4211±213 km$^2$ of which 2119 km$^2$ remained in 2003 (-50%
or -0.32% a$^{-1}$) and 1806 km$^2$ in 2015 (-57% or -0.35% a$^{-1}$). This is a loss of 313 km$^2$ or 15% (-1.2% a$^{-1}$) for P2. In the eastern
Alps (regions 9-14) the relative area loss for P3 is -64% compared to -52% for the western Alps. Highest area losses are found
in the Cottian and Maritime Alps (Region 2) with -92.5%, Dolomites and Carnic Alps (Region 13) with -82% and the Lepontine
Alps (Region 8) with -78%. The least affected regions are the Pennine Alps (Regions 5) with -45% and the Bernese Alps
(Region 6) with -44% (cf. Table 2 and Figure 3a, the changes per country are listed in Table S4). At least to some extent, the
larger glaciers in Regions 5 and 6 caused the smaller relative area changes, but in absolute terms, they are higher ( Figure 3a).
The size dependency is also reflected by the glacier area changes per size class, where small glaciers have higher relative area
losses than large glaciers (Figure S11). Glaciers smaller than 1 km$^2$ (in 1850) lost 74% of their area until 2015 whereas glaciers
between 5 and 10 km$^2$ lost 46% and the two glaciers larger than 50 km$^2$ lost 20% of their area.
For P2, the total glacier area shrunk by 15% (-1.22% a$^{-1}$), but many of the mostly very small glaciers (287) had a larger area
in 2015 than in 2003. This is caused by differences in interpretation from different analysts, sensor resolutions (Landsat vs.
Sentinel-2) and mapping conditions (snow, clouds and shadow) rather than by growing glaciers. The given 2003 to 2015 area
change rate should thus be considered as a lower bound, as correcting the 2015 outlines to the 2003 interpretation would have
led to an even larger area loss.

### 3.2 Glacier elevation changes

Glaciers in the entire Alps experienced severe volume loss since the LIA (Figure 3d). The mean elevation change for P3 over
the entire Alps was -43.3 m without a significant difference between the eastern and the western Alps. The highest changes
were observed in the Eastern (Region 10; -50.2 m) and southern Rhaetian Alps (Region 11; -46.7 m) and the Bernese Alps
(Region 6; -46.4 m). Generally, elevation changes were largest at an elevation of around 1600 m (dominated by Region 6) and
decreasing towards higher elevations (Figure 4). The smaller elevation changes at low elevations can be explained by the
smaller ice thickness during the LIA. The largest elevation changes (-105 m) were found at 1650 m in the western (Figure 4a)





and at 2250 m (-65 m) in the eastern Alps (Figure 4b), basically reflecting the larger glaciers reaching further down in the western Alps. In elevations between 2150 and 3950 m, elevation changes were very similar in the eastern and western Alps.

**Figure 3: Glacier change measures averaged per sub-region for a) to c) and as a raster product for d)- The panels show: a) Relative area changes [%] in relation to total LIA area, d) elevation changes, b) volume changes [%] in relation to total LIA volume, c) acceleration of volume change rates for P1 compared to P2 (Hugonnet et al., 2021). All background images: ESRI, (2023a).**

### 3.3    Glacier volume changes

The total glacier volume of the Alps at their LIA maximum extent is calculated as 281±43 km$^3$ of which 99.6±12.6 km$^3$ remained in 2015 (-65%). Considering the uncertainty (15.3%) and a possible underestimation due to missing glaciers of 5%, the LIA volume could be as high as 338.4 km$^3$ and as low as 238 km$^3$. Thereby, the western Alps lost 105.8±9 km$^3$ (-58.5%), whereas the eastern Alps lost 75.6±6.4 km$^3$ (-75.3%). The total volume change was highest in regions 3, 5, and 6 (western Alps) as well as 10 and 12 (eastern Alps), i.e. the regions with the largest glaciers (Figure 3a). Relative volume change was most dramatic in regions 1 (-75.5%), 2 (-96.8%), 4 (-75.1%) and 8 (-81.9%) in the western Alps and regions 12 (-79.9%), 13 (-88.9%) and 14 (-78.2%) in the eastern Alps, i.e. apart from Region 12 those with the smallest glaciers (Figure 3b; values



per country are listed in Table S4). Overall, volume change was highest in an altitude between 2500 m and 3000 m (Figure
S2), i.e. the elevation range with the largest area. This compensates for the lower mean elevation change at this altitude. 3D
hillshade visualizations of the LIA and modern glacier surface can be seen in Figures S13-S16.

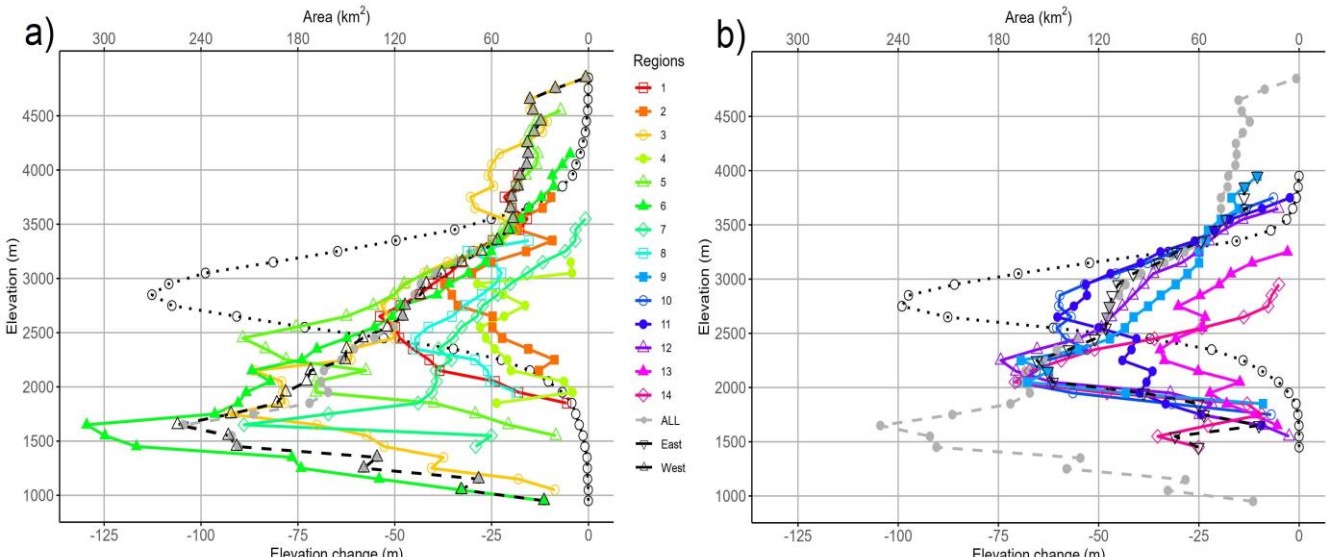


**Figure 4: Elevation changes with elevation per sub-region for a) the western Alps and b) the eastern Alps. The regional means are
shown in black and the mean of the entire Alps are in grey. The black dotted line indicates the LIA area for the specific elevation
band.**

### 3.4 Increase in change rates

Area, elevation, and volume change rates were much higher in P2 compared to P1. The glacier area change rate was nearly
four times higher for P2 (-1.23% a⁻¹) compared to P1 (-0.32% a¹) (Table 2, Figure S3). Thereby, the increase in the western
Alps (4.9x) is more than two times larger compared to the eastern Alps (2.4x). In Region 12 (Tauern Alps West), the area
change rates for P2 almost didn't change, beyond mapping uncertainties. In Region 4 (Savoy Prealps) fast melting glaciers led
to the largest area change rate increase (11.9x), whereas Region 6 (Bernese Alps) experienced the lowest area change rate until
2000 (-0.22% a⁻¹) but is also showing a strong increase (6.1x).
Overall, elevation change rates were 3.3 times higher for P2 as derived by Hugonnet et al. (2021) compared to P1. Here, the
increase was a bit larger in the western (3.5x) than in the eastern Alps (3.0x). Regionally, the increase was largest in Regions
7 (5.7x), 4 (5x), 8 (4.5x) and 9 (4.2x) (Figure 3c). The change is also dependent on the elevation with the elevation loss rate
decreasing towards higher elevations (Figures S4 and S5). Notable is the small increase in Region 13, which could be explained
by the presence of mostly small glaciers (partly only remnants left) with short response times that now experience only small
changes. When calculating the change rates for P2 with the data from Sommer et al. (2020) and the DEM difference between
the COP DEM and the SRTM DEM (Figures S6 and S7), the regional variability is similar, but the increase in the elevation
change rate is lower compared to the dataset from Hugonnet et al. (2021). Further research is necessary to investigate what



causes the differences among the available datasets. More detailed views of elevation change patterns before and after the year
2000 are shown in Figures 5 and S10.
The absolute volume change rates increased by 38% in P2 compared to P1. Interestingly, whereas the western Alps experienced
a strong increase in the volume change (48%), the eastern Alps experienced only a slight increase (14%). Nevertheless, some
regions have shown a lower volume loss rate for P2 compared to P1 (Regions 2, 12, 13 and 14). The volume change rates for
larger river basins increased by 54%, for the Rhône and 55% for the Rhine. The other basins have about constant volume loss
rates, even slightly decreasing after 2000 (-17%) in the south-eastern Alps (Adige, Piave, Brenta, Tagliamento and Soča). A
table of country and basin-specific area and volume changes can be found in Tables S4 and S5.

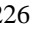
**Figure 5: Examples of elevation change rates between the LIA and 2000 (a and c) and 2000-2014 after Hugonnet et al. (2021) (b and**
**d) for the Bernese Alps (a and b) and the Ötztal Alps (c and d) using the same colour legend for both periods. All background images:**
**ESRI, (2023a).**





**Table 2: Glacier area, and volume elevation changes for each region as well as total areas and volumes. Also listed are long term and recent change rates. P1 stands for the period between LIA and around 2000, similarly P2 for 2000-2015 and P3 for LIA-2015. Elevation change rate for P2 taken from Hugonnet et al. (2021).**

| Region ID | Main division | Region name | Area LIA (km²) | Area 2003 (km²) | Area 2015 (km²) | Rel. area change P1 (%) | Rel. area change P3 (%) | Rel. area change rate P1 (% a⁻¹) | Rel. area change rate P2 (% a⁻¹) | Volume LIA (km³) | Volume 2015 (km³) | Volume change P3 (km³) | Mean elev. change P3 (m) | Elev. change rate P1 (m a⁻¹) | Elev. change rate P3 (m a⁻¹) | Elev. change rate P2 (m a⁻¹) | Increase rate P2/P1 | Change of median elevation P3 (m) |
|---|---|---|---|---|---|---|---|---|---|---|---|---|---|---|---|---|---|---|
| 1 | west | Dauphiné Alps | 159.04 | 90.77 | 64.76 | -42.93 | -59.28 | -0.28 | -2.39 | 7.98 | 1.96 | -6.02 | -40.2 | -0.24 | -0.24 | -0.81 | 3.42 | 125 |
| 2 | west | Cottian & Maritime Alps | 20.69 | 2.68 | 1.55 | -87.03 | -92.52 | -0.57 | -3.53 | 0.58 | 0.02 | -0.56 | -30.63 | -0.15 | -0.19 | -0.42 | 2.73 | 23 |
| 3 | west | Graian Alps | 636.44 | 332.27 | 267.42 | -47.79 | -57.98 | -0.31 | -1.63 | 43.35 | 15.76 | -27.6 | -44.22 | -0.25 | -0.27 | -0.74 | 2.99 | 105 |
| 4 | west | Savoy Prealps | 16.45 | 8.25 | 4.4 | -49.84 | -73.24 | -0.33 | -3.89 | 0.51 | 0.13 | -0.39 | -22.74 | -0.14 | -0.14 | -0.69 | 5.01 | 76 |
| 5 | west | Pennine Alps | 703.03 | 431 | 387.67 | -38.69 | -44.86 | -0.25 | -0.84 | 50.31 | 21.51 | -28.8 | -41.7 | -0.24 | -0.25 | -0.73 | 3.02 | 158 |
| 6 | west | Bernese Alps | 689.78 | 462.42 | 389.37 | -32.96 | -43.55 | -0.22 | -1.32 | 64.48 | 31.68 | -32.8 | -46.38 | -0.26 | -0.28 | -1.03 | 3.88 | 128 |
| 7 | west | Glarus Alps | 107.52 | 53.23 | 41.49 | -50.5 | -61.41 | -0.33 | -1.84 | 5.91 | 2.46 | -3.44 | -30.07 | -0.15 | -0.18 | -0.86 | 5.71 | 98 |
| 8 | west | Lepontine Alps | 182.72 | 52.4 | 39.51 | -71.32 | -78.37 | -0.47 | -2.05 | 7.51 | 1.36 | -6.15 | -33.84 | -0.18 | -0.21 | -0.8 | 4.52 | 144 |
| 9 | east | Rhaetian Alps West | 352.62 | 147.96 | 118.38 | -58.04 | -66.43 | -0.38 | -1.67 | 18.53 | 5.52 | -13.01 | -37.48 | -0.19 | -0.23 | -0.82 | 4.19 | 135 |
| 10 | east | Rhaetian Alps East | 470.85 | 207.99 | 185.97 | -55.83 | -60.5 | -0.36 | -0.88 | 31.41 | 7.87 | -23.53 | -50.18 | -0.29 | -0.3 | -0.82 | 2.78 | 81 |
| 11 | east | Rhaetian Alps South | 284.48 | 122.53 | 100.59 | -56.93 | -64.64 | -0.37 | -1.49 | 18.06 | 4.9 | -13.16 | -46.67 | -0.28 | -0.28 | -0.95 | 3.43 | 110 |
| 12 | east | Tauern Alps West | 540.64 | 195.52 | 194.76 | -63.83 | -63.98 | -0.42 | -0.03 | 30.72 | 6.18 | -24.54 | -45.51 | -0.28 | -0.28 | -0.61 | 2.22 | 138 |
| 13 | east | Dolomites & Carnic Alps | 23.24 | 4.8 | 4.2 | -79.35 | -81.91 | -0.52 | -1.03 | 0.61 | 0.07 | -0.54 | -26.5 | -0.15 | -0.16 | -0.24 | 1.58 | 69 |
| 14 | east | Northeastern Alps | 23.62 | 7.32 | 6.17 | -69 | -73.89 | -0.45 | -1.32 | 1 | 0.22 | -0.78 | -32.41 | -0.21 | -0.2 | -0.45 | 2.14 | 37 |
| 15 | west | Western Alps | 2515.67 | 1433.02 | 1196.17 | -43.06 | -52.45 | -0.28 | -1.38 | 180.63 | 74.88 | -105.76 | -42.26 | -0.24 | -0.26 | -0.84 | 3.51 | 147 |
| 16 | east | Eastern Alps | 1695.45 | 686.12 | 610.07 | -59.53 | -64.02 | -0.39 | -0.92 | 100.31 | 24.76 | -75.56 | -44.89 | -0.26 | -0.27 | -0.78 | 2.96 | 134 |
| 17 | All | Alps | 4211.12 | 2119.14 | 1806.24 | -49.97 | -57.27 | -0.32 | -1.23 | 280.95 | 99.64 | -181.31 | -43.32 | -0.25 | -0.26 | -0.82 | 3.3 | 143 |




### 3.5 Glaciers that melted away

Temperature increase has caused at least 1832 glaciers with a LIA area of 292 km$^2$ to melt away. This is a lower bound estimate
because several glaciers that were not mapped in 2003 or 2015 were also not mapped with their LIA extent. Most of these
glaciers can be found in Regions 5 and 6 (Pennine and Bernese Alps) with the largest area loss in Regions 3 and 9 (Graian and
Rhaetian Alps West). These regional differences have uncertainties because different analysts have likely worked along a
different rule set for the mapping and might thus not have included all disappeared glaciers (this also applies to the newly
digitised LIA glaciers). Nevertheless, some previously glacierized catchments like the Val Chamuera in the Engadin
(Switzerland) are now basically ice-free (Figure S8).

### 3.6 Change in topographic parameters

The median glacier elevation, which can be used as a proxy for the balanced-budget ELA$_0$ (Braithwaite and Raper, 2009),
increased from 2897 m during the LIA to 3040 m in 2015 (+143 m). The western Alps experienced a slightly higher increase
(147 m) than the eastern Alps (134 m). The change was largest in the Pennine Alps (Region 5; 158 m) and the Lepontine Alps
(Region 8; 144 m). The smallest changes were observed in the Cottian and Maritime Alps (Region 2; 23 m) and the North-
eastern Alps (Region 14; 37 m).

## 4 Discussion

### 4.1 Influence of methods on glacier volume change and comparison with other studies

Our estimate of the LIA glacier area is 263.2 km$^2$ (6%) smaller than the value estimated by Zemp et al. (2008) and thus outside
our uncertainty range, even if considering already disappeared and not digitised glaciers. It could thus be that the extrapolation
method applied by Zemp et al. (2008) gives slightly too large areas for the LIA. This is reasonable when considering that area
change rates have recently strongly increased. Applying them backwards would result in too large areas with this method.
Comparing the reconstructed volumes with the GIS-based method applied here with values calculated with the
parameterisation scheme by Haeberli and Hoelzle (1995), a large difference is visible. In total, the parameterisation scheme
results in a 25% lower total glacier volume for the LIA (224 km$^3$ vs. 281±43 km$^3$ in our study). This is also visible on a regional
scale where the parameterisation scheme is lower in all but three regions (9, 13, and 14). Especially Regions 3 and 6, where
some of the largest glaciers in the Alps are located, had 43% and 25% lower volumes with the parameterisations scheme.
However, for 2015 the volume differences are only 1.2 km$^3$ (or 1.2%) smaller with the parameterisation scheme (99.6±12.6 vs
98.4 km$^3$). Although this would lead to the conclusion that the GIS-based surface reconstruction overestimates LIA glacier
volumes, we speculate that the approach by Haeberli and Hoelzle (1995) rather underestimates LIA volumes. For example,
the mean slope of the glaciers might have increased so that mean glacier thickness decreased. It also needs to be considered
that the parameterisation scheme has its limitations and works best if glacier extents are in balance with climatic conditions
(which is certainly not the case in 2015). When the GIS-based surface reconstruction overestimates glacier volumes, this also
applies to the calculated volume change rates and the recent acceleration of volume loss rates found here would be even larger.





Looking at specific glaciers with long observation periods, GLAMOS (2022) published volume changes starting from 1850-
1900 (digitised from historic maps) for some Swiss glaciers. For most of these, volume changes are in good agreement with
our estimated (e.g. Great Aletsch Glacier: -6.8 km$^3$ (1880-2017), vs -6.6 km$^3$ in P3). Some outliers exist, for example, the
Lower Grindelwald glacier. Here, GLAMOS (2022) estimated the volume change between 1861 and 2012 to be -0.44 km$^3$,
whereas our calculations resulted in -1.2 km$^3$ and the parameterisation scheme in -0.57 km$^3$. The Lower Grindelwald glacier
is one glacier where the bi-linear elevation change gradient could not be calculated due to the low correlation between elevation
and elevation change rate, thus the surface was only reconstructed using the outline points, leading to an overestimation of the
LIA surface elevation, especially in the (comparably large) accumulation area. However, as the differences could be positive
or negative, we would argue that at the granularity of the regional aggregation shown in Figure 1 and Table 2, the volume
changes obtained here are likely very accurate (within 5% of the real value), but at the scale of individual glaciers deviations
might reach 50% or more, depending on the specific characteristics of a glacier.
The difference in the LIA volumes between the parameterisation scheme and the GIS-based reconstruction increases with
increasing glacier area and decreases with mean slope (Figure S9). Therefore, for large, flat glaciers like those found in Regions
3 and 6, the difference is greatest. The parameterisation scheme uses only mean slope (derived from glacier length and elevation
range) to determine mean ice thickness and might thus underestimate volumes for large and bottom-heavy glaciers such as
Aletsch, Unteraar or Gorner where a large part of the volume is stored in the lower, flat part. Also, Lüthi et al. (2010) found
that the volume at the end of the LIA was larger relative to the length of the glaciers, confirming that the parameterisation
scheme might underestimate glacier volume. The parameterisation scheme by Haeberli and Hoelzle (1995) might thus provide
a minimum estimate of LIA glacier volumes.
On the other hand, the parameterisation scheme and GIS-based reconstruction gave very similar results for the thickness change
rate. The mean elevation change rate for P3 using the GIS-based reconstruction is -0.26 m a$^{-1}$ whereas it is -0.25 m a$^{-1}$ with the
parameterisation scheme. Regionally, the difference between the methods can be much larger, with the rate from the GIS-
based method being 45% higher in Region 14 (-0.36 m a$^{-1}$ vs -0.2 m a$^{-1}$) and 29% lower in Region 6 (-0.22 m a$^{-1}$ vs. 0.28 m a$^{-1}$)
compared to the parameterisation scheme (Figure 6). Results published by Hoelzle et al. (2003) also using the parameterisation
scheme are in line with our results, giving -0.11 m w.e. a$^{-1}$ for small glaciers and -0.25 m w.e. a$^{-1}$ for large glaciers between
1850 and 1996 for the Swiss Alps.



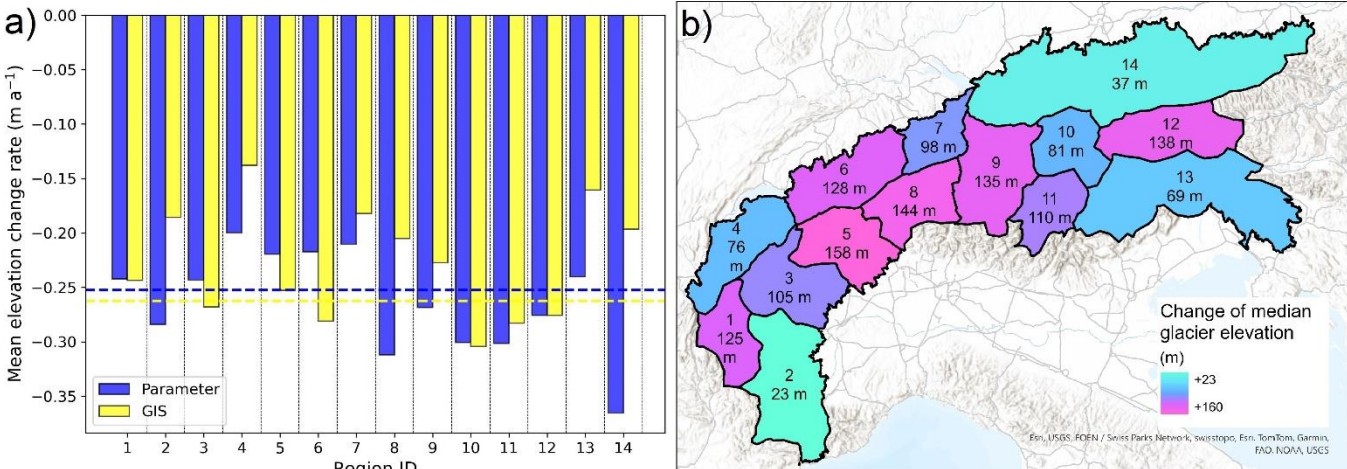

**Figure 6: a) Mean elevation change rate for each region (calculated from parameterisation and GIS approach). Dashed lines indicate the Alpine-wide mean rate b) Region ID and regional change of the median glacier elevation between the LIA and 2015. Background image: ESRI, (2023a).**

## 4.2 Influence of timing on glacier change rates

The change rates since the LIA also depend on the date of the LIA maximum. Since this is a bit different for each glacier and only known for some of them, an approximate regional average of 1850 has been used. To assess the impact of the LIA maximum date on the calculated change rates, a 20-year upper and lower bound was applied. The area change rates would decrease from -0.35% $a^{-1}$ for 1850 to -0.31% $a^{-1}$ when using 1830 and increase to -0.39% $a^{-1}$ when starting in 1870. Similarly, the elevation change rates would decrease from -0.26 m $a^{-1}$ to -0.23 m $a^{-1}$ and -0.3 m $a^{-1}$, respectively. Thereby, the impact of the LIA starting date on elevation change rates is not linear but increases towards a smaller date range ( Figure S12).

Finally, since P1 is much longer than P2, the rates have to be interpreted with caution. Between the LIA maximum and the year 2000 most glaciers in the Alps experienced at least two periods with glacier stagnation or even re-advances (1920s and 1980s), which results in a lower overall change rate compared to a period with a constant decrease, i.e. glaciers in the Alps were basically retreating and losing mass continuously since the year 2000.

## 4.3 Climatic and hydrological implications

The observed change in median elevation of 143 m would translate to a temperature increase of 0.84 to 1.43 °C, depending on the atmospheric lapse rate applied (Haeberli et al., 2019; Kuhn, 1989; Rolland, 2003; Zemp et al., 2007). This is lower than the 1.5° and 1.6° temperature increase determined by Begert and Frei (2018) and Auer et al. (2007) for Switzerland and the Alps, respectively. Precipitation trends since the 19[th] century are inconclusive, but the Alpine region has become somewhat drier and sunnier since the 1990s (Auer et al., 2007), both enhancing glacier melt. However, as glaciers are not in balance with the current climate, their ablation regions will continue shrinking and thus shifting the median elevation further up-wards. For the large glaciers with flat tongues, this effect is somewhat compensated by the ongoing surface lowering.



The impact of long-term ice loss extends beyond the immediate glacierized landscape, affecting glacier runoff and water availability. The excess melt (imbalance) of the glacier adds to the overall runoff with its usual seasonal variations. Our calculations reveal that the absolute volume loss rate in the eastern Alps has only slightly increased in P2 (14%), indicating that the peak of the imbalance contribution is near. Indeed, some regions in the eastern Alps (Regions 12-14) experienced a decreasing imbalance contribution, implying that peak water in those regions might have occurred already. Moreover, the rivers in the south-eastern Alps flowing into the Adriatic Sea also experienced a decreasing glacier imbalance contribution and the basins draining into the Po and Danube rivers showed stagnating volume loss rates, indicating that peak water might be reached in the near future. Similarly, Huss and Hock (2018) suggest that European basins may have already reached or be on the brink of reaching peak water. On the other hand, the volume loss rates continued to increase dramatically until at least 2015 in the western Alps (except in Region 2). Nevertheless, according to Huss et al. (2008) the peak run-off in highly glacierized basins in the western Alps will be reached in the coming decades.

## 5    Conclusion

This study has calculated the massive glacier area and volume loss in the European Alps since the end of the Little Ice Age. After the compilation of existing and manual digitising of missing LIA glacier outlines, we obtained a 99% coverage by area. For these glaciers, the total area was 57% smaller in 2015 (1806 km$^2$) compared to the LIA maximum (4211±213 km$^2$). The LIA glacier surface reconstruction with a GIS-based approach resulted in an estimated volume loss of 181±15.5 km$^3$ or 65% of the original 281±43 km$^3$. Despite the strongly reduced glacier area by the year 2003, the post-2000 period (P2) witnessed about three times higher rates of elevation loss than in the mean for the LIA to 2000 period (P1), indicating an increasing impact of climate forcing. At the same time, the run-off contribution by glacier imbalance was decreasing after 2000 in some regions of the eastern Alps, while still increasing in the western Alps.

Due to the temperature increase, at least 1832 glaciers melted away, with numerous others diminished to small remnants of their previous extent. The median glacier elevation was 143 m higher in 2015 than at the end of the LIA and will further increase, as most glaciers have not yet adjusted their geometry to current climatic conditions. The resulting deglaciation of entire mountain catchments with related effects on the Alpine landscape will thus also continue. This has far-reaching implications for water resources, run-off, ecosystems, hydropower production and tourism in the Alpine region and requires timely consideration. The here presented dataset will certainly help in assessing the impacts of climate change on mountain landscapes in further detail.

**Competing interests**
The contact author has declared that none of the authors has any competing interests.





**Acknowledgement**

The work of J. R. is supported by PROTECT. This project has received funding from the European Union's Horizon 2020

research and innovation programme under grant agreement No 869304, PROTECT contribution number XX. The work of F.P.

has been performed in the framework of the ESA project Glaciers_cci+ (4000127593/19/I-NB).

**Authors contributions**

J. R. led the study and the writing of the paper and performed the glacier surface reconstruction as well as all data analysis. F.

P. provided ideas and comments and contributed to the writing of the paper and the digitising of outlines.

**Data availability statement**

LIA surface elevations and new outlines will be made available using an online repository (Zenodo). The new LIA outlines

will also be available from the GILIMS glacier database.



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
