# Peer review of "Reconstructed glacier area and volume changes in the European Alps since the Little Ice Age"

_EGUsphere, 2024_

## Referee Comment (RC2)

**Review of 'Reconstructed glacier area and volume changes in the European Alps since the Little Ice Age'**
by Johannes Reinthaler and Frank Paul

The manuscript presents a study of past glacier area, volume, and bed reconstruction for the European Alps, utilizing geomorphological feature identification and interpretation on high-resolution images, combined with historical topographic maps and current glacier boundaries. The results show that past glaciers have shrunk by over 50% in aerial extent, with more than 60% of their total volume lost between 1850 and 2015, which is consistent with a few previous studies in the region. Additionally, the authors estimate surface elevation of the LIA glacier extent (using an interpolation method they developed) and estimated elevation changes of the glaciers during the study period, and reported a tripling of thinning rates after 2000 compared to the overall long-term thinning rate.

In my opinion, the manuscript reconstructs and presents a crucial dataset of past glacier extents, which is vital for understanding long-term glacier behaviour in the Alps. The insights inferred from previous glacier fluctuations will also be invaluable for future glacier response modelling in the region. Considering the very high rates of glacier retreat and thinning in the Alps in the current era of climate change, understanding past rates is essential for planning mitigation strategies. Additionally, these datasets would be valuable to a range of scientific disciplines beyond glaciology, including hydrology, geohazard management, ecology, and others.

Overall, the manuscript is well-written in its different sections and concise in nature. The authors missed a few important references, as pointed out by community scientists, and they have promised to include these in the revised version. I do not have any major criticism. Below, I outline a few general comments and several minor suggestions for improvement. I recommend a minor to moderate level of revision before the manuscript's final online publication.

If any of my comments are unclear, please do not hesitate to contact me for further clarification.

**General comments**

1. I did not find any discussion on the contrasting area/thickness/volume loss between the LIA and the current period in the western (high loss) and eastern Alps (low loss) regions. While the authors briefly touched on overall climatic conditions, there isn't a dedicated discussion on this topic, which would be valuable for understanding the influence of climatic shifts in these two contrasting (high/low loss) areas. If the authors do not plan to present a dedicated discussion, at least adding a few lines addressing climate changes in these regions, along with some basic statistics, would give readers a brief understanding of the influence of climate and its spatial variation across the European Alps, thus better establishing the connection between climate and glacier loss.

2. The authors might consider using the total uncertainty propagation technique from Mannerfelt et al. (2022). Reviewer 2 has raised concerns about the uncertainty estimation of volume changes in the current work, and I concur with these concerns. Mannerfelt et al. (2022; equation 9 in their work) used a similar dataset covering LIA outlines and varying temporal periods, which I think would provide valuable insights/guides for addressing the uncertainty issues in the current study.

**Line-by-line comments**

L14: Here, I would have mentioned the latest year instead of 'today,' as in future years, the term 'today' will lose its relevance.

L17: I would prefer 'On average' (which sounds better I think!) instead of 'In the mean'?

L32: 'trim lines' and trimlines (in abstract; L11; also in L40, and elsewhere) needs to be consistent across the manuscript. Also, I think, as the authors have already expanded LIA in the abstract, they should not need to re-expand it. Please check with the journal's guidelines.

L39-40: I think this sentence is incomplete. Did the authors mean '....between 1350 and 1850/60, with the exact timing depending on the glacier"? Please check.

L78-80: Here, I would be happier to see 'the range of resolutions' for the very-high-resolution images those were used for interpretation/delineation.

L82-83: Which study provides the outlines from 1967-1971 for France? The authors might consider citing the reference here for the readers' quick knowledge.

L103-105: This modern DEM refer to 10 m Copernicus DEM? Right. Re-mentioning might be helpful for the readers.

L105-106: Is this 'Topo to Raster' a tool for interpolation or a known method? Not clear from the current sentence? Please clarify.

L107-108: The authors mean 'The output result..', right?

L153: Please expand SLR here, for the readers' sake.

L176-177: Here, please mention the range of the elevation changes, for further information to readers. Also, here, 'highest changes' sounds a bit awkward as the value is the most negative (lowest), so I would suggest something like 'highest mass loss/thinning'?

**Table**

Table 1: I would suggest the authors to mention the time period of P1-P2 for clear information in the table caption.

**Figures**

Figure 3: Are the volume change rasters (subplot d) aggregated to a specific grid size? If so, please mention it.

Figure 5: By looking at the panel b and the color contrast of elevation changes, it seems the change values are even lower than -2 m/y, if I am not wrong. The authors need to re-draw the colorbar or add extend marks.

Figure 6: Here, in panel b, the color scheme of the colorbar may not be ideal, given the range of +23 to +123 (only increasing side everywhere). A single color gradient, where the darkest shade represents the highest increase and the lighter shades represent the lowest values, might be more appropriate.

Figure S6:  I would suggest the authors to label the sub-plots by citations (e.g., Sommer et al., 2020, etc.) and by the time period of the elevation change estimate. Also, same in and S10.

**References**

Mannerfelt, E. S., Dehecq, A., Hugonnet, R., Hodel, E., Huss, M., Bauder, A., and Farinotti, D.: Halving of Swiss glacier volume since 1931 observed from terrestrial image photogrammetry, The Cryosphere, 16, 3249–3268, https://doi.org/10.5194/tc-16-3249-2022, 2022.

---

## Community Comment (CC1)

La Grave

▲ La Meije
m

Saint Christophe-
en-Oisans

▲ Barre des Écrins
m

Glacier de Mont de Lans
Glacier de la Girose
Glacier du Tabuchet
Glacier de la Selle
Glacier d'Arsine
Glacier de la Plate des Agneaux
Glacier de Bonne Pierre
Glacier Blanc
Glacier Noir
Glacier de la Pilatte
Glacier du Chardon
Glacier des Rouies
Glacier du Vallon des Etages
Glacier des Aupillous
Glacier de Chabournéou
Glacier Est des Sellettes
Glacier du vallon de Lanchâtra
Glacier du Sélé
Glacier de Séguret Foran
Glacier de l'Aup
Glacier de Faravel

▲ L'Olan
m

▲ Les Bans
m

Vallouise

La Chapelle-en-
Valgaudemar

▲ Le Sirac
m km

---

## Author Comment (AC2)

**Response to the comments by M. Le Roy (CC1)**

Dear,

First, congrats to the authors for this long-awaited and well conducted work!

Thank you very much!

I have two comments on the manuscript:

**1) Regarding Western Alps outlines**

The Figure 2 shows Ecrins massif with 'new' outlines in 'blue'.

However, these LIA glacier outlines were already available (please see the figure attached here) in Marie Gardent's 2014 PhD thesis (where the 'yellow' outlines you use originate I suppose)

So why have you redigitized them and presented them as new here (Fig. 2)?

By the way, the two Gardent's references listed at the end are not actually quoted anywhere in the main text/figures of the manuscript.

So, why not quoting them in the caption of Figure 2, as it seems yellow outlines come from them?

If not, please provide a reference here for previously available outlines.

Thank you very much for pointing out to the datasets created as part of Marie Gardent's PhD thesis. The yellow outlines in Figure 2 were actually downloaded through the Glariskalp database, but the link is not working anymore (http://www.glariskalp.eu/?it_inventario-delle-estensioni-attuali-e-passate-dei-ghiacciai,9). For these outlines, we have cited the Gardent and Deline (2011) study (with a spelling error), as this appeared to be the contribution of France to the Glariskalp project. Back then, we missed to check if the datasets presented in the PhD thesis of Marie Gardent (2014) were also digitally available and decided to map the missing regions ourself. We now asked and got access to the full dataset and used it instead of our outlines as it is more complete. We will thus recalculate area and volume changes for this region using the LIA outlines of Marie Gardent, cite her in the analyst field of the attribute table in the merged alpine-wide dataset and will also cite the PhD thesis in the main text.

Regarding the two references, in L34 we have cited Garent and Deline (2011) with a spelling error (missing d) which we will change to Gardent (2014). We will also acknowledge Antoine Rabatel, who has kindly provided the complete dataset to us.

**2) Regarding the timing of LIA maximum extent**

It sounds to me somehow misleading and too simplistic to present differences in the timing of LIA maxima as depending only on the geographical location (e.g Western Alps/Italy in 1820 CE vs Austria in 1850 CE) as it was done here in the Introduction of the manuscript.

There might be a misunderstanding here. We do actually not make a difference in the LIA maximum timing and have used 1850 to calculate change rates for all glaciers (see L118). The values presented in the introduction should just give an overview of selected studies presenting results on LIA maximum timing. Additionally, in Section 4.2 we describe the effects of the LIA timing on change rates. We will also add the references mentioned below for a clearer presentation of the large regional variability.

Better, the response time of glaciers should be mentioned as one of the (most likely) explanations for the differences in the chronology of LIA glacial maxima.

Indeed, we showed recently in two review papers dealing with the early LIA period (https://journals.sagepub.com/doi/full/10.1177/09596836221088247) and the Holocene (https://www.sciencedirect.com/science/article/abs/pii/B9780323997126000180?via%3Dihu) in the Alps, that (small) glaciers reached LIA maximum extent during any periods of LIA glacier maxima. For instance, many small glaciers, located from the Ecrins massif (to the west) up to the Tirol (to the east), reached absolute LIA maxima during the early 14[th] century.

We agree that the timing of the maximum extent might have a relation to glacier response times, but (in contrast to some other regions) all earlier LIA moraines can be found very close to the 1850 outlines (see Figure 20.10 in Chapter 20 of the cited book). Using an earlier date would thus result in a misleading calculation of area and volume change rates. However, to not confuse it in the text with the real maximum extent, we will refer to it as the late LIA maximum extent instead of the LIA maximum position.

The LIA chronology is therefore biased towards large (and most slowly reacting) glaciers because majority of available dates come from these sites, where most work has been carrying out in the past.

Yes, we agree that this could be the case. However. there are also exceptions from the rule. With a view on the study by Nussbaumer and Zumbühl (2011), the quickly responding Oberer Grindelwald Glacier had its final maximum extent also after 1850, the position just a bit shorter than in 1820 and 1600. And the slowly reacting Mer de Glace has the same number of maximum extent peaks since 1600 (about 8) as the fastly reacting Glacier de Bossons. So response times are likely only one possible factor for differences in the chronology.

Related comment for Line 42:

*'in contrast to other regions in the world, extent differences (e.g. between 1850, 1820 or 1600) are small in the Alps'*

Please provide references to support your point here. This affirmation does not seem straightforward to me at all.

We will add some references for this statement. For example, the comparison shown in Fig. 2 of the PAGES news (Vol. 19, No 2, Page 69) from 2011 and the study by Nussbaumer and Zumbühl mentioned above should make the point clear: Glacier maxima for the Alps all reach about the same extent between 1600 and 1860 (cf. also Fig. 20.10 in the book chapter), whereas

for some outlet glaciers of the Jostedalsbreen Ice Cap in Norway there is a steady decrease after they reached their maximum extent around 1750. Similarly for many glaciers in the Andes with their multiple and progressively younger moraine walls (e.g. Rabatel et al., 2008). Even if not precisely dated everywhere, these well-separated moraine sequences indicate rather different response characteristic compared to the Alps where the 1850 extents where about as large as those before. Hence, for the Andes change rate calculations have to consider the longer period when using the outermost moraine wall for the extent (see also Reinthaler and Paul, 2023).

Mainly because:

Even if it's relatively true at large Alpine glaciers, it is less at small glaciers.

But foremost, dates with temporal resolution high enough to assert this (archives, tree rings) are barely available outside the Alps

Yes, the sample of properly dated LIA moraines is indeed still small, but we think in the meantime we have some good data from many regions in the world. However, for the statement above one might not even need the dating. There are typical regional differences in moraine deposition which allow us using the '1850' extent as a proxy for the LIA maximum extent in the Alps, but this is not possible for certain glaciers in southern Norway or many glaciers the Andes.

Using 1850 as an averaged reference date to calculate glacier area and volume change rates for glaciers in the Alps is certainly a simplification, but we think sufficient for a first estimtion at regional scales. As in similar regional-scale LIA studies (e.g. Carrivick et al. 2023 https://doi.org/10.1029/2023GL103950), we provide lower and upper bounds to consider the uncertainty (and variability) of the timing.

We are looking forward to the many studies that will further refine and improve the LIA glacier outline dataset presented here.

---

## Author Comment (AC3)

**Response to the comments by R. R. Colucci,(CC3)**

I read this work with interest, as it certainly represents a necessary contribution to the better understanding of glacial evolution in perhaps the most studied mountain sector in the world when dealing with mountain glaciers. Nonetheless, my view is that the uncertainties of assessment that a work of this type inevitably entails could have been minimized by wider cooperation. In a mountain chain as deeply studied as the Alps, the individual specificities of the various Alpine sectors are certainly clearer locally.

Thank you for your comments. We fully agree that such a work is best be done by working together and have thus reached out to several colleagues who described datasets in publications, but have not shared the digital data (shape files) so far. We also decided to publish the study in The Cyosphere as its open discussion potentially allowed identifying even more datasets. This worked very well. It is also clear that this study could only be a starting point. Further improvements can likely be applied at the scale of individual glaciers and we hope that colleagues will contact us in the case our outlines need to be revised.

After this general comment, I will go into the specifics of the area of my major competence. Particularly, on line 163 and in several figures, I may disagree with the subdivision of area 13, as it would be more correct to indicate Dolomites and Julian Alps, instead of Carnic Alps. In fact, in the Carnic Alps there is only one documented glacier in Austria, the Eiskar Glacier.

Thank you for this suggestion. As the Eiskar Glacier in the Carnic Alps is quite famous (also due to its isolation) we would like to keep the name but add the Julian Alps. So region 13 is now named 'Dolomites, Carnic & Julian Alps'.

I might suggest also reading and citing the following as it represents the first detailed inventory of the LIA Julian Alps glaciers with the first details on what in this preprint is discussed at paragraphs 3.1, and 4

Thank you for the note. So far we have cited the study Colucci and Zebre (2016) for the dataset you provided. There is no problem to change the citation to the ESPL paper mentioned below, if this is the correct reference. In any case, we are happy to add the results of the area changes in our discussion section.

Colucci R.R. (2016). Geomorphic influence on small glacier response to post Little Ice Age climate warming: Julian Alps, Europe. Earth Surface Processes and Landforms, 41: 1227-1240

---

## Author Comment (AC4)

**Replies to the review comments by the anonymous reviewer on 'Reconstructed glacier area and volume changes in the European Alps since the Little Ice Age'.**

by Johannes Reinthaler and Frank Paul

The manuscript by Reinthaler and Paul describes a glacier surface and extent reconstruction for the Little Ice Age maximum (assumed to be in 1850) and calculates area, thickness and volume changes from the LIA maximum to 2000 and 2015. The study makes use of numerous existing datasets and newly determines LIA outlines and surface topography for 19% of all glaciers in the Alps ($>0.1$ km$^2$) for which outlines were missing in previous inventories. I find that the study uses generally robust methods to generate novel and significant findings. The manuscript is well-structured and well-written, the presentation is clear, and the results, analysis and discussion provide useful perspectives on acceleration of changes as well as uncertainties. I suggest the manuscript can be published after minor revisions. My comments are detailed below. I have some doubts about the method that is used to calculate the total volume error estimate, which I would like to see addressed. Other (minor) comments are mostly suggestions for textual revisions, added discussion, clarification and additional references.

Dear reviewer, thank you very much for taking the time to carefully read our manuscript. We are happy to read that the study has a robust method and that the analysis was done well. Below, we go through your comments and address each of them.

Comments:

Volume uncertainty: The current error estimate of the volume does not seem correctly calculated. Summing squared relative errors and taking the square root is normally applied to calculate the total error of a variable that is the product (or division) of other variables with known errors (see e.g. https://www.statisticshowto.com/statistics-basics/error-propagation/). The total volume is however not the product of glacier outlines, surface reconstruction, and bed topography, but rather the product of the related ice thickness and area. Therefore, a better way to calculate volume errors is to apply error propagation on the product of area and thickness where relative errors of glacier area and thickness are individually defined. I think this formula could be used; eps_V = V*sqrt((eps_H/H)^2 + (eps_A/A)^2) where eps_V, eps_A and eps_H are volume (V), area (A) and thickness (H) errors, respectively.

Thank you for the suggestion! You are right that the volume uncertainty is in the end related to uncertainties in glacier area and ice thickness, but in our study ice thickness is already a derived variable resulting from the independently derived glacier beds (which we are just using) and the glacier surface (which we have reconstructed). So, in our case we have to consider the uncertainties of glacier outlines (i.e. area) reconstructed by us, the uncertainty in elevation of the reconstructed glacier surfaces (which we also have calculated) and the uncertainty in the elevation of modelled bed topography (which we take from the related publications). As these are the three independent sources of uncertainty to be considered, we think our way of calculating volume uncertainty is correct.

L19-20: "Many glaciers have now only remnants of their former coverage left, which led to deglaciation of entire catchments.". Please specify how many glaciers this applies to and which catchments are affected, because now it is a rather hollow statement.

We agree that the statement is not quantitative, but think this is sufficient for the abstract as a detailed specification would be rather lengthy and is presented in the main text. It touches upon the difficult question when a glacier is gone. This might be defined by size or when there is no flow. For size, the widely used threshold is 0.01 km2; but what if three or more pieces smaller than this but forming a remaining ice body larger than 0.01 km2, are left? We would prefer to not enter into this discussion in the abstract and prefer to stay with the more generalized statement.

L20: "The new datasets should support a wide range of studies related to the effects of climate change in the Alps.". This is also rather vague. It could be good to specify types of studies / study fields that would benefit from the dataset.

We agree, but think this is a rather long list. We have now written: The new datasets should support a wide range of studies related to the determination of climate change impacts in the Alps, e.g. future glacier evolution, model validation, hydrology, surface albedo and land cover change, plant succession and emerging hazards.

L36-37: References and links to GLIMS and RGI should be included here and/or in the Data Availability section.

We have added the references for GLIMS and RGI.

L40-41: "However, in contrast to other regions in the world, extent differences (e.g. between 1850, 1820 or 1600) are small in the Alps.". Is there a reference for this?

We have listed references for example in the following sentences. We do not cite any general study since this is our conclusion reviewing relevant literature.

L51-54: "Whereas reconstructions ... European Alps.". This could be merged with / moved to the paragraph ending at L38.

We have moved the paragraph to the desired location.

L76-77: "Due to differences in interpretation of glacier extents by different analysts for the two datasets, we will only present glacier changes at a regional scale rather than per glacier.". This helps in case the observer errors are random, but not when they are systematic. Are there any indications for potential systematic errors between the two outline inventories?

Only regionally. For example, some regions in Austria had less good snow conditions in 2016 than in 2003 and included many of the perennial ice and snow fields, leading to additional 'glaciers' being mapped for the 2016 inventory. There is also a random difference in interpretation due to the different sensor resolutions. The 10 m Sentinel-2 images show details that are not visible at 30 m resolution and might have been digitized larger or smaller in 2003.

L82: "As a starting point for the LIA outline digitizing, we used outlines from 1967-1971 (for France) and the RGI v7.0 from 2003 for the other regions.". What is meant by "as a starting point" here? Is it just to get a rough idea where the glacier is? Or is more information extracted than that?

"As a starting point" means that they were used as a base before modifying them to fit the LIA moraines. We have rephrased the sentence.

L95-96: "A few glaciers that melted away before 2003 would lower this number by a few decimals.". What data is this statement based on? Is it possible to give an estimate of the number of glaciers that may have melted entirely since the LIA maximum?

The statement is not based on specific datasets, but the general observation that glaciers also decreased in size between the LIA and 2003. Hence, without outlines in 2003 (or 1967-1971 for France), we cannot reconstruct former glacier extents and thus also not determine the number of glaciers that completely melted away. This would require having a 'perfect glacier map' from the LIA that is consistent with today's interpretation. Nevertheless, we have rephrased the sentence to make it clearer.

L102-103: "The method calculates a scaling factor by dividing the gradient by the LIA elevation change (from interpolating outline points only)." This is not very clear. Please give a short summary of this method (as described in Reinthaler and Paul, in review) and how it compares to other methods.

We agree and have added a few more details about the method, but to keep the text short we would prefer to refer to the Reinthaler and Paul (2024) publication for a full description (which is now published).

L105: Please add a reference for the Topo to Raster toolbox.

We have added this reference: Hutchinson, M. F.: A new procedure for gridding elevation and stream line data with automatic removal of spurious pits, J. Hydrol., 106, 211–232, https://doi.org/10.1016/0022-1694(89)90073-5, 1989.

L118-119: "Similarly, we have used the year 1850 as the date of maximum LIA extent.". This is a (justifiable) assumption of the timing of LIA maximum extent. It would be good to add this information already in the abstract or introduction.

In the abstract we write "around 1850" to indicate that this date has a bit of variability. We hope this is sufficient here.

L128: "This probably resulted in positive elevation changes in several accumulation areas (Figure S6).". Does this also explain differences with Sommer et al. (2020)?

Yes, this could be one of the reasons. However, we have not performed a more detailed analysis on this as it is beyond the scope of this study.

L144-145: See my previous comment about possibly incorrect quantification of the volume error.

See above, we think our assessment is correct. We have, however, exchanged 'glacier outlines' with 'glacier area' to be clearer.

L170: "15%". Is it coincidence that this is exactly the same value as the volume loss or is it inherent to the method that was used to reconstuct the surface shape back in time which preserves much of the modern glacier shape (i.e. area - thickness distribution)?

As both values were determined completely independent and also the input datasets were (slightly) different, we would say the coincidence is by chance. Please note, the changes reported here for the period P2 are not related to our surface reconstruction, but are based on area change rates calculated from subtracting the 2015/16 from the 2003 area and volume change rates from Hugonnet et al. (2021).

L171-172: "This is caused by differences in interpretation from different analysts, sensor resolutions (Landsat vs. Sentinel-2) and mapping conditions (snow, clouds and shadow) rather than by growing glaciers.". Is there any reference for this? I also wonder whether a similar overestimation in 2015 likely also applies to the other (larger) glaciers?

We have described the problem in Sections 4.4 and 6 as well as Fig. 10 of Paul et al (2020) which has been added. It is mostly related to smaller glaciers located at higher elevations (and being impacted by remaining seasonal snow and topographic shadow).

L179-180: "Generally, elevation changes were largest at an elevation of around 1600 m (dominated by Region 6) and decreasing towards higher elevations (Figure 4).". Can this elevation also be calculated for P2 (2000-2015)? With glaciers retreating I would expect this elevation to migrate upwards as well with time.

Yes, this is possible. For P2 the elevation with the highest elevation change is at 1750 m. We have added this value to the manuscript.

L181-182: "The largest elevation changes (-105 m) were found at 1650 m in the western (Figure 4a) and at 2250 m (-65 m) in the eastern Alps (Figure 4b)". This does not seem consistent with the 1600 m for the entire Alps (?).

The reason why the value for the entire Alps is very similar to the one for the western Alps, is because no glacier in the eastern Alps reaches down to the elevation of 1600 m. So the 'signal' is dominated by the western Alps.

Figure 3: In the caption please indicate which period it applies to (P1 or P3).

The values correspond to P3. We have edited the figure caption accordingly.

L191: "238 km$^3$". Why is the 5% for the missing glaciers only added to the upper bound and not to this lower bound?

This is because missing glaciers could only correspond to an underestimation rather than overestimation, i.e. they introduce a bias.

Figure 4: This figure could be improved. The resolution is currently rather poor, reducing readability. Furthermore, the right axis is missing in both panels.

This is indeed a very busy figure. We now added the missing right border. The resolution of the image will be improved for the final version.

L210-212: "Overall, ... (Figure 3c).". Since these glaciers are not experiencing frontal ablation, elevation change rates, and trends of those, could be compared with longterm surface mass balance trends. I am not well up to date regarding available long-term SMB datasets for the Alps, but when available it would be interesting to compare long-term SMB trends with the surface height change trends presented here.

Yes, fully agreed but it comes along with several caveats as study periods and glacier extents used will differ (e.g. extents are adjusted each year for field-based SMB measurements). We have, however, presented some comparisons in the discussion section.

L216-217: "the regional variability ... Hugonnet et al. (2021).". It could be good to indicate the average relative changes compared to P1 for these as well.

Yes, this can be done. We have added the related mean elevation change rates for the DEM differencing and Sommer et al. (2020) as well as Hugonnet et al. (2021) datasets.

L217-218: "Further research is necessary to investigate what causes the differences among the available datasets". It could be considered to include a comparison with surface mass balance studies, e.g. Davaze et al. (2020; https://doi.org/10.3389/feart.2020.00149), in order to have some independent source to

compare to. I would also like to see some more discussion on potential causes of the discrepancies between the available datasets.

Yes, this would be great but doing this carefully would require a separate study resulting in a long additional paper (one of the many studies we mention now at the end of the abstract). A related analysis is thus beyond the scope of this study where we focus on glacier changes since the LIA.

L220: "The absolute volume change rates...". I suppose this is based on the use of Hugonnet et al. (2021) for P2? It could be good to clarify since in the previous paragraph different surface height change datasets were considered.

Yes, the volume change rates are here calculated from the Hugonnet et al. (2021) dataset. We have now added this information.

L265: "which is certainly not the case in 2015". I may be missing the point here. Why are 2015 conditions relevant for the Haeberli and Hoelze estimate (published in 1995)?

We only refer here to the method of Haeberli and Hoelze (1995), not the estimate. Their method works best when glaciers are close to a dynamic equilibrium state, which is not the case for glaciers in the European Alps in 2015.

L277: "depending on the specific characteristics of a glacier". It could be interesting to discuss these characteristics a bit further (as it might give some directions for future work).

Yes, this would be one of the mentioned further interesting studies that can now be performed. As a related investigation is beyond the scope of this study. we now cite Reinthaler and Paul (2024) who look at such characteristics a bit closer. Apart from this, we think that the scientific community will have a large number of further ideas for follow-up studies.

L300: "an approximate regional average of 1850 has been used.". Larger response times of large (thick) glaciers will likely cause many of those glaciers to have a later LIA maximum extent date than small glaciers. Although this may be hard to account for in the approach, at least some discussion on the size dependence would be good.

We agree, but would again enter here into uncharted territory, i.e. requiring another study that can now be performed. Regarding response times, we think that slope is more important than size and the considerable overlap of slope classes for differently sized glaciers would result in an unclear relation with glacier size. We also note that the response of smaller glaciers might be increasingly impacted by non-climatic controls, e.g. due to shading their evolution can be largely decoupled from climatic changes. Despite their small size, their response times might get very long. Some discussion of the impact of the maximum extent timing and calculated area change rates can be found in Reinthaler and Paul (2023) which we have now also cited.

L310-313: "The observed change in median elevation of 143 m would translate to a temperature increase of 0.84 to 1.43 °C, depending on the atmospheric lapse rate applied (Haeberli et al., 2019; Kuhn, 1989; Rolland, 2003; Zemp et al., 2007). This is lower than the 1.5° and 1.6° temperature increase determined by Begert and Frei (2018) and Auer et al. (2007) for Switzerland and the Alps, respectively.". This needs some clarification. I assume the temperature increase in the first sentence is purely due to the elevation drop. But what is the 1.5-1.6 deg C change in the second sentence referring to? Does it include both anthropogenic warming and elevation-drop induced warming?

Yes, the first 4 studies are cited for the range of possible atmospheric lapse rates. These are required to transform the change of median elevation into a change of temperature. The other two studies only refer

to the observed atmospheric warming rather than vertical lapse rates, providing an independent dataset for comparison not related to glacier changes.

L317-327: Interesting discussion on peak runoff!

Thank you.

Conclusions: The conclusions section gives a good summary of the work. However, I do miss some recommendations for future work. Would there for example be room for inverse (ice flow) modelling methods to generate physically more robust LIA surface topography?

Thank you. As mentioned in the abstract and being the reason for the vague statement of possible future applications, there is a near endless number of possibilities for future work and we think the community will have their own ideas. We would thus prefer to not enforce specific applications here.

Textual revisions:

All text errors have been corrected

L10 (and elsewhere): digitising --> digitization
L17: In the mean --> On average
L18: 1600 m --> 1600 m above sea level (m a.s.l.)
L55: "first" --> "first complete"
L60: "Regional subdivision" --> "Study regions"
L99: "DEM" --> "Digital Elevation Model (DEM)"
L112: "the total glacier volume" --> "the contemporary total glacier volume"
L141: "Uncertainties of the bed topography impact" --> The impact of bed topography uncertainty"
L143: Add "volume" between "overall" and "uncertainty".
L160: "reconstructed to" --> "estimated at"
L170: "shrunk" --> "shrank"
L196: "altitude" --> "altitude range"
L201: "are" --> "is"
L203: "Increase in change rates". Please change to something more specific (indicating the variables it refers to).
L252: "if considering" --> "when including"
L261: "would" --> "could"
L269: "estimated" --> "estimate"
Figure 6b: Please mention that changes are negative.
L330: "coverage by area" --> "areal coverage"
L331: "these glaciers" --> "all glaciers" (otherwise it refers to the missing glaciers)
L356: "GILIMS" --> "GLIMS"

---

## Author Comment (AC5)

**Replies to the review comments by Arindan Mandal on 'Reconstructed glacier area and volume changes in the European Alps since the Little Ice Age'.**

by Johannes Reinthaler and Frank Paul

The manuscript presents a study of past glacier area, volume, and bed reconstruction for the European Alps, utilizing geomorphological feature identification and interpretation on high-resolution images, combined with historical topographic maps and current glacier boundaries. The results show that past glaciers have shrunk by over 50% in aerial extent, with more than 60% of their total volume lost between 1850 and 2015, which is consistent with a few previous studies in the region. Additionally, the authors estimate surface elevation of the LIA glacier extent (using an interpolation method they developed) and estimated elevation changes of the glaciers during the study period, and reported a tripling of thinning rates after 2000 compared to the overall long-term thinning rate.

In my opinion, the manuscript reconstructs and presents a crucial dataset of past glacier extents, which is vital for understanding long-term glacier behaviour in the Alps. The insights inferred from previous glacier fluctuations will also be invaluable for future glacier response modelling in the region. Considering the very high rates of glacier retreat and thinning in the Alps in the current era of climate change, understanding past rates is essential for planning mitigation strategies. Additionally, these datasets would be valuable to a range of scientific disciplines beyond glaciology, including hydrology, geohazard management, ecology, and others.

Overall, the manuscript is well-written in its different sections and concise in nature. The authors missed a few important references, as pointed out by community scientists, and they have promised to include these in the revised version. I do not have any major criticism. Below, I outline a few general comments and several minor suggestions for improvement. I recommend a minor to moderate level of revision before the manuscript's final online publication.

If any of my comments are unclear, please do not hesitate to contact me for further clarification.

Thank you very much for the constructive review of our manuscript. It is nice to hear that the study is valued for its contribution to the understanding of long-term glacier changes.

**General comments**

I did not find any discussion on the contrasting area/thickness/volume loss between the LIA and the current period in the western (high loss) and eastern Alps (low loss) regions. While the authors briefly touched on overall climatic conditions, there isn't a dedicated discussion on this topic, which would be valuable for understanding the influence of climatic shifts in these two contrasting (high/low loss) areas. If the authors do not plan to present a dedicated discussion, at least adding a few lines addressing climate changes in these regions, along with some basic statistics, would give readers a brief understanding of the influence of climate and its spatial variation across the European Alps, thus better establishing the connection between climate and glacier loss.

We have actually not discussed climatic conditions, as the observed differences in area/volume changes are largely dependent on glacier size (partly also hypsometry) and making a connection to regional variability in climate change would be very speculative. We would also prefer not to discuss climate here in detail as we have not investigated this and think that the details of regional variability are rather complicated (e.g. doi.org/10.1002/joc.1377), i.e. beyond the scope of this study. We would thus prefer to stay with the data we have derived rather than including more speculative parts. We will, however, add some more details on the differences in glacier changes between eastern and western Alps in the discussion.

The authors might consider using the total uncertainty propagation technique from Mannerfelt et al. (2022). Reviewer 2 has raised concerns about the uncertainty estimation of volume changes in the current work, and I concur with these concerns. Mannerfelt et al. (2022; equation 9 in their work) used a similar dataset covering LIA outlines and varying temporal periods, which I think would provide valuable insights/guides for addressing the uncertainty issues in the current study.

We appreciate the recommendation, but think that our uncertainty estimation is fitting better to our datasets, since we have used independent datasets from three specific sources from which we know their specific uncertainties. Please see also our response to Reviewer 1: "Thank you for the suggestion! You are right that the volume uncertainty is in the end related to uncertainties in glacier area and ice thickness, but in our study ice thickness is already a derived variable resulting from the independently derived glacier beds (which we are just using) and the glacier surface (which we have reconstructed). So, in our case we have to consider the uncertainties of glacier

outlines (i.e. area) reconstructed by us, the uncertainty in elevation of the reconstructed glacier surfaces (which we also have calculated) and the uncertainty in the elevation of modelled bed topography (which we take from the related publications). As these are the three independent sources of uncertainty to be considered, we think our way of calculating volume uncertainty is correct.

**Line-by-line comments**

L14: Here, I would have mentioned the latest year instead of 'today,' as in future years, the term 'today' will lose its relevance.
We wrote now 'around 2015' instead of 'today'.

L17: I would prefer 'On average' (which sounds better I think!) instead of 'In the mean'?
Done.

L32: 'trim lines' and trimlines (in abstract; L11; also in L40, and elsewhere) needs to be consistent across the manuscript. Also, I think, as the authors have already expanded LIAin the abstract, they should not need to re-expand it. Please check with the journal's guidelines.
We have checked the consistency of trimlines vs trim lines and now use trim.... All acronyms have to be repeated in the main text as required by the journal guidelines.

L39-40: I think this sentence is incomplete. Did the authors mean '            between 1350 and 1850/60, with the exact timing depending on the glacier"? Please check.
Correct, thank you for pointing this out. We have changed the text accordingly.

L78-80: Here, I would be happier to see 'the range of resolutions' for the very-high-resolution images those were used for interpretation/delineation.
We have added the resolution.

L82-83: Which study provides the outlines from 1967-1971 for France? The authors mightconsider citing the reference here for the readers' quick knowledge.
We have now added the reference.

L103-105: This modern DEM refer to 10 m Copernicus DEM? Right. Re-mentioning mightbe helpful for the readers.
Yes, agreed. We have slightly rephrased the sentence.

L105-106: Is this 'Topo to Raster' a tool for interpolation or a known method? Not clearfrom the current sentence? Please clarify.
It is the name of a tool in the ESRI software package. We have now clarified it and included a reference.

L107-108: The authors mean 'The output result..', right?
Yes, indeed. Changed.

L153: Please expand SLR here, for the readers' sake.
We have changed it.

L176-177: Here, please mention the range of the elevation changes, for further informationto readers. Also, here, 'highest changes' sounds a bit awkward as the value is the most negative (lowest), so I would suggest something like 'highest mass loss/thinning'?
We have added elevation change details and changed 'highest changes' to 'highest thinning'.

**Table**

Table 1: I would suggest the authors to mention the time period of P1-P2 for clearinformation in the table caption.
We have added the information to the table caption.

**Figures**

Figure 3: Are the volume change rasters (subplot d) aggregated to a specific grid size? Ifso, please mention it.
We have added the grid size (4 km).

Figure 5: By looking at the panel b and the color contrast of elevation changes, it seems the change values are

even lower than -2 m/y, if I am not wrong. The authors need to re-draw the colorbar or add extend marks.
We have changed the colour bar labels since also values smaller than -2 all have the same colour.

Figure 6: Here, in panel b, the color scheme of the colorbar may not be ideal, given the range of +23 to +123 (only increasing side everywhere). A single color gradient, where the darkest shade represents the highest increase and the lighter shades represent the lowest values, might be more appropriate.
Yes, agreed. We have changed the colour bar.

Figure S6: I would suggest the authors to label the sub-plots by citations (e.g., Sommer etal., 2020, etc.) and by the time period of the elevation change estimate. Also, same in andS10.
Yes, this could be done, but we think the information is now also well accessible. We would thus prefer to keep the two figures as they are.

**References**

Mannerfelt, E. S., Dehecq, A., Hugonnet, R., Hodel, E., Huss, M., Bauder, A., and Farinotti,D.: Halving of Swiss glacier volume since 1931 observed from terrestrial image photogrammetry, The Cryosphere, 16, 3249–3268, https://doi.org/10.5194/tc-16-3249-2022, 2022.

---

## Author Response (AR1)

Dear Johannes Reinthaler and Frank Paul,

thank you for sketching a detailed plan of action on how to address the various community and review comments. I was most delighted about the contributions in the public discussion pointing to other existing and valuable literature or data sources. Thank you for considering them.

Thank you very much. We also think that including all available data is helping in increasing the overall quality of the dataset.

Apart from that I sense that multiple major comments remain unresolved or at least controversial. These relate to the timing of the LIA in the European Alps, quantification of volume uncertainties, comparison with existing long-term SMB studies, etc. Moreover, the authors did unfortunately overlook to answer my initial comments - some of which redundant with the reviewer comments. In addition, the elevation values, where highest elevation changes are found, appear inconsistent. The Alpine-wide value is smaller than the respective values in the eastern and western parts. Finally, please refer to existing literature on glacier response times (e.g., Zekollari).

Thank you for pointing out the points that still need further clarification.

LIA timing:

Concerning the timing of the LIA, we think that due to the complex nature and variability of timing over an entire region, and the already available literature reviews (Le Roy et al., 2024) we decided to keep the text short and focus on the overall trend with some examples. However, we have cited now the study by Le Roy et al. (2024) to let the readers explore this theme in more depth if desired. In the revised ms we make clear that some glaciers were much earlier also slightly larger so that we do not analyse maximum extents, but the latest possibly only near-maximum extent. The impact of a 20-year uncertainty in the timing on the calculated area change rates is also given. As mentioned before, our study will not solve all problems related to the reconstruction of former glacier extents and should rather be seen as a starting point for further investigations.

Reference dataset/long term SMB:

We have now compared our elevation and volume changes for selected regions with the results from Mannerfelt et al. (2022). Although differences in the datasets used (outlines, DEMs) create larger deviations, our results can now be seen in relation to a dataset from an additional time-step, i.e the evolution of change rates can be better followed. We emphasize that this is only a rough first-order estimate, unsuitable for wider implications.

Highest elevation change values:

We have also checked the elevation change values. We previously provided first a rounded value of 1600 m and in the next sentence the correct value of 1650 m. This is indeed confusing and we will remove the rounded number. Also, in response to a suggestion of reviewer 2, we added the elevation with the highest elevation change for P2 for both the eastern and western Alps.

Volume uncertainty:

Regarding the volume uncertainty, we understand the reasoning behind the suggested formula eps_V = V*sqrt((eps_H/H)^2 + (eps_A/A)^2). However, the volume uncertainty is in our case not only linked to ice thickness/bedrock and area uncertainties, but also includes the surface reconstruction uncertainty. These three are variables that we have used as input and for which we have (independent) uncertainty estimates. We have now clarified the different sources of uncertainty in the text and hope

our method for the uncertainty calculation is sufficient considering the scope and scale of our dataset. Since we are analysing regional glacier changes, a simplified uncertainty estimation rather than a glacier specific or even pixel-based estimation seems more fitting and we would thus leave the calculation as is.

Overlooked initial comments

Lastly, regarding answering the initial comments, we had posted our answers already on the 2nd of May 2024. Please have a look and let us know if anything else should be revised.

In summary, I invite you to submit a revised version of your manuscript addressing the above pending points. To clarify these points, I suggest a second review round.

Although we could not implement all suggestions, we hope that the performed adjustment and explanations in the responses are satisfactory. We thank the editor and all reviewers for their careful reading and constructive suggestions.

---

## Author Response (AR2)

We thank the editor J. Fürst for the management of the review process and the constructive feedback. We hope that all points made by the two reviewers have now been resolved and that the manuscript can be published.

Response to reviewer comments

Reviewer 1:

I thank the authors for revising the manuscript. Nearly all my previous comments have been satisfactorily addressed. I only have two minor remarks (below) and recommend acceptance of the manuscript after these have been addressed.

Thanks you very much for the constructive feedback. We have addressed now the remaining point.

Section 2.5 (Volume uncertainty). The formula used for calculating the relative volume error is correct as it is. It could be useful though to include in the manuscript how the formula for the relative volume error is derived. It is based on the underlying equation where $V = A*H = A*(s-b)$ where s is surface height, b is bed height and A is area. Error propagation of the product $(A*H)$ gives $eps\_V/V = sqrt((eps\_A/A)^2+(eps\_H/Hmean)^2)$. Furthermore, error propagation of the subtraction (s-b) gives $eps\_H^2 = eps\_b^2 + eps\_s^2$. Combining these gives $eps\_V/V = sqrt((eps\_A/A)^2 + (eps\_s/Hmean)^2 + (eps\_b/Hmean)^2))$, which as it seems is the equation that was used by the authors.

Thank you very much for the further explanation of this formula. We have revised the text in the manuscript and supplement and integrated the suggested details of explanation. We also added information about the different uncertainty components and their origin. We hope it is now clearer.

L218-220: "The total glacier volume of the Alps at their LIA maximum extent is calculated as 280±43 km3 of which 99.6±12.6 km3 remained in 2015 (-64%). Considering the uncertainty (15.3%) and a possible underestimation due to missing glaciers of 4.8%, the LIA volume could be as high as 336 km3 and as low as 237 km3."

This may need some further revision. In case there is indeed a likely underestimation of LIA volume because of missing glaciers, then the correct approach (I think) would be to simply add the 4.8% to the 280 km3, i.e. 293 km3 would be the best LIA volume guess. The ±43 km3 error bounds would then approximately apply to the new volume estimate, but it would be even more correct to add uncertainty in the missing glacier volume to the previous 43 km3 uncertainty estimate. The combined error could be calculated using $sqrt(43^2+X^2)$ where X is the error (in km3) for the missing glaciers.

Thank you for the suggestion to also consider the volume of the missing glaciers in the total. You are right that including this part would give a more realistic value, but in our impression this component is so uncertain, that we would prefer to not include it and keep the mean LIA volume value as is. We have added this information to the main text and supplement to make our choice clearer.

Reviewer 2 (Arindan Mandal):

The authors have satisfactorily addressed my previous concerns and incorporated the suggested or necessary changes. The current version of the manuscript is well-written, and I do not have any further major concerns, aside from a few minor or technical corrections that should be addressed before publication.

*Thank you very much, we are happy to hear that all the important points have been amended satisfactorily.*

Line numbers are according to the revised manuscript file (without track-change version).

L15-17: I think '(-57%)' should be after '4244 km2' as it is like the authors did for volume loss in the second part of the sentence.

*We have rewritten the sentence.*

L48-50: Can you re-check the sentence because it sounds a bit awkward in its current form. The authors may change something like: 'In the Alps, glaciers reached their maximum extents several times between 1250 and 1850/60, with the exact timing varying by glacier.'

*We have rewritten the sentence.*

L58-60: Here, I would suggest using decimal numbers consistently. 2271.6 → 2272 km2

*Done.*

L104-106: Please remove 'the' in front of 'glaciers'

*Done.*

L110: The glacier → Glaciers

*Done.*

L247: Please expand COP DEM here, I understand it is Copernicus DEM, but it would better for the readers.

*Done.*

L270: I would suggest changing 'analysts' to 'studies'

*Done.*

L309-311: Here, 'namely' is not fitting/sounding well because after 'namely' values are there. Instead, they may change it to something like: '....after 1931, with -29.4 km³ and -3.8 km³ reported by Mannerfelt et al. (2022), compared to -32.8 km³ and -3.5 km³ observed in this study.'

*We have rewritten the sentence.*

L311: 'rate' and 'values' in this case is same, so please remove any of it.

*Done.*

L387-388: For volume change uncertainty values, please be consistent with the decimal numbers.

*We have checked the uncertainty values and decided not to include decimals for consistency.*

Figures S4: What is the unit of the acceleration rate (legend colorbar), please add it in the colorbar for easy comprehension, like as it in the Fig S6/S10.

*We have explained the meaning of the acceleration rate in the caption. Since it's a relative increase factor (e.g. threefold increase) it has no physical unit.*